# Usefulness of KL-6 for Predicting Clinical Outcomes in Hospitalized COVID-19 Patients

**DOI:** 10.3390/medicina58101317

**Published:** 2022-09-21

**Authors:** Mikyoung Park, Mina Hur, Hanah Kim, Chae Hoon Lee, Jong Ho Lee, Minjeong Nam

**Affiliations:** 1Department of Laboratory Medicine, Eunpyeong St. Mary’s Hospital, College of Medicine, The Catholic University of Korea, Seoul 03312, Korea; 2Department of Laboratory Medicine, Konkuk University School of Medicine, Seoul 05030, Korea; 3Department of Laboratory Medicine, Yeungnam University College of Medicine, Daegu 42415, Korea; 4Department of Laboratory Medicine, Korea University Anam Hospital, Seoul 02841, Korea

**Keywords:** KL-6, prediction, clinical outcome, COVID-19, biomarker

## Abstract

**Background**: Krebs von den Lungen 6 (KL-6) is a novel biomarker for interstitial lung disease, and it reflects acute lung injury. We explored the usefulness of KL-6 to predict clinical outcomes in hospitalized coronavirus disease 2019 (COVID-19) patients. **Methods**: In a total of 48 hospitalized COVID-19 patients, KL-6 levels were measured using the HISCL KL-6 assay (Sysmex, Kobe, Japan) with the HISCL 5000 automated analyzer (Sysmex). Clinical outcomes (intensive care unit [ICU] admission, ventilator use, extracorporeal membrane oxygenation [ECMO] use, and 30-day mortality) were analyzed according to KL-6 percentiles. Age, initial KL-6 level, Charlson comorbidity index (CCI), and critical disease were compared using the receiver operating characteristic (ROC) curve and Kaplan-Meier methods for clinical outcomes. **Results**: KL-6 quartiles were associated with ICU admission, ventilator use, and ECMO use (all *p* < 0.05), except 30-day mortality (*p* = 0.187). On ROC curve analysis, initial KL-6 level predicted ICU admission, ventilator use, and ECMO use significantly better than age, CCI, and critical disease (all *p* < 0.05); age, initial KL-6 level, CCI, and critical disease predicted 30-day mortality comparably. On Kaplan–Meier survival analysis, hazard ratios (95% confidence interval) were 4.8 (1.2–19.3) for age, 4.7 (1.1–21.6) for initial KL-6 level, 3.9 (0.9–16.2) for CCI, and 2.1 (0.5–10.3) for critical disease. **Conclusions**: This study demonstrated that KL-6 could be a useful biomarker to predict clinical outcomes in hospitalized COVID-19 patients. KL-6 may contribute to identifying COVID-19 patients requiring critical care, including ICU admission and ventilator and/or ECMO use.

## 1. Introduction

The coronavirus disease 2019 (COVID-19) pandemic, caused by severe acute respiratory syndrome coronavirus 2 (SARS-CoV-2), has still continued to threaten the public since it broke out in Wuhan in 2019 [1,2,3]. The clinical course of COVID-19 shows a broad spectrum from mild to critical disease [4]. Critical COVID-19 patients, especially those with acute respiratory distress syndrome (ARDS), could progress to an acute crisis requiring intensive care unit (ICU) admission, mechanical ventilation (MV), and extracorporeal membrane oxygenation (ECMO) [3,4,5,6]. It is known that up to 31% of COVID-19 patients could present COVID-19-induced ARDS, and it may progress into pulmonary fibrosis (PF) as a post-acute COVID-19 syndrome [7].

Angiotensin-converting enzyme (ACE) 2 plays a central role in the pathogenesis of COVID-19 [8,9,10,11,12]. When SARS-CoV-2 enters host cells via ACE2, angiotensin (Ang) II is produced, followed by the production of proinflammatory cytokines, such as tumor necrosis factor-alpha (TNF-α) and interleukin-6 (IL-6) [8,9,10,11,12]. It results in inflammation, fibrosis, lung damage, and edema. Hyper-inflammation by SARS-CoV-2 induces fibroblast activation, increased extracellular matrix, and collagen deposition leading to PF, similar to the features of interstitial lung disease (ILD) [13,14]. It is known that risk factors for PF include old age, comorbidities, ICU admission, and MV [7,13]. Age and comorbidity are major factors influencing the prognosis of COVID-19, and they are included in several COVID-19 prognostic models [3,15]. Of the two, age showed better prognostic performance [16]. Both adjusted odds ratio (OR), hazard ratio (HR), and the predictive point of age over 70 years were higher compared with sex, body mass index, symptoms, and comorbidity [16]. The above results of comorbidity were heterogeneous according to its type [16]. Meanwhile, the predictive performance of the Charlson comorbidity index (CCI) above three, an estimated comorbidity burden, was similar to that of age in predicting severe/critical COVID-19 [17].

KL-6 is a novel biomarker for ILD, and it is produced by injured/regenerating type II pneumocytes as well as bronchial epithelial cells and other cells [7,18,19]. Recent studies have explored the association between KL-6 levels and COVID-19 disease severity or prognosis; KL-6 levels increased according to COVID-19 disease severity and significantly predicted hospital days and poor prognosis in some studies [20,21,22,23,24,25,26,27,28,29,30,31,32]. KL-6 has been also suggested as a potential biomarker for post-COVID-19 PF [7]. KL-6 was significantly associated with radiological abnormalities after COVID-19 and predicted post-COVID-19 PF [22,30].

To the best of our knowledge, no studies have evaluated the predictive performance of KL-6 compared with age, comorbidities, and disease severity in COVID-19 simultaneously. We hypothesized that KL-6 might be beneficial in predicting clinical outcomes in COVID-19. We explored the usefulness of KL-6 for predicting clinical outcomes in hospitalized COVID-19 patients compared with age, CCI, and WHO disease severity. We also analyzed serial KL-6 levels according to the 30-day mortality.

## 2. Materials and Methods

### 2.1. Study Population

The enrollment and clinical outcomes of the study population are presented in Figure 1. From February to June 2020, a total of 48 COVID-19 patients derived from a previous study were enrolled [2]. The enrollment criteria were as follows: hospitalized adult patients over 20 years of age with available serial residual ethylene-diamine-tetraacetic acid (EDTA) plasma samples after routine blood tests and known 30-day status. Medical records were reviewed thoroughly to assess demographic, clinical, and laboratory data. CCI, the sequential organ failure assessment (SOFA), and WHO disease severity were assessed at enrollment as described previously [3,33,34]. Clinical outcomes included ICU admission, ventilator use, ECMO use, and 30-day mortality.

In the total study population, moderate and critical disease was observed in 27.1% (*n* = 13) and 72.9% (*n* = 35), respectively, with no mild or severe disease. In critical disease patients, sepsis and septic shock were 82.9% (*n* = 29) and 17.1% (*n* = 6), respectively. Among the 13 patients with moderate disease, 1 patient was admitted to ICU. Among the 35 patients with critical disease, 12 patients were admitted to ICU. Among them, ventilators were applied to 11 patients (ARDS [*n* = 9]), and ECMO was applied to 6 ventilated ARDS patients. The 30-day mortality was 16.7% (*n* = 8).

### 2.2. KL-6 Assay

A total of 332 residual EDTA plasma samples were consecutively collected from 48 patients from enrollment to discharge or death. The collected residual samples were aliquoted into 200 μL and were stored at –70 °C until measurement. Frozen samples were thawed at room temperature and gently mixed immediately before measuring KL-6 levels. KL-6 levels were measured using the HISCL KL-6 assay (Sysmex, Kobe, Japan) with the HISCL 5000 automated analyzer (Sysmex) based on a two-step sandwich chemiluminescence enzyme immunoassay.

The manufacturer’s reference interval was up to 398.0 U/mL. Analytical measurement intervals were from 10 to 6000 U/mL. KL-6 levels were measured according to the manufacturer’s instructions. Finally, 96 consecutive KL-6 levels from 48 patients were included for statistical analysis. Initial KL-6 (*n* = 48) indicated KL-6 level at enrollment. Follow-up (F/U) KL-6 (*n* = 48) indicated KL-6 level at discharge in survivors or at death in non-survivors.

### 2.3. Statistical Analysis

Data were presented as a number (percentage) or median (interquartile range, IQR). The Shapiro–Wilk test was used for assessing the normality of data distribution. The Kruskal–Wallis test and chi-squared test were used to compare the four groups according to KL-6 quartiles (from Q1 to Q4). KL-6 quartiles were as follows; Q1 < 160.0 U/mL, 160.0 U/mL ≤ Q2 < 234.5 U/mL, 234.5 U/mL ≤ Q3 < 449.0 U/mL, and Q4 ≥ 449.0 U/mL.

With the receiver operating characteristic (ROC) curve analysis, the optimal cut-off value of age, CCI, and WHO disease severity was 70 years, 3, and critical disease, respectively. The optimal cut-off value of initial KL-6 was 412 U/mL for ICU admission, ventilator use, and ECMO use, which was the same for all, and 322 U/mL for 30-day mortality. The distribution of age > 70 years, initial KL-6 level > 412 U/mL, CCI > 3, and critical disease was compared according to ICU admission, ventilator use, and ECMO use, using the chi-squared test or Fisher’s exact test. For 30-day mortality, initial KL-6 level > 322 U/mL was applied. Mann–Whitney test or Wilcoxon signed-rank test were used to compare KL-6 levels between moderate and critical diseases, between sepsis and septic shock, and between survivors and non-survivors according to the 30-day mortality.

In the ROC curve analysis, AUC, sensitivity, and specificity of age > 70 years, initial KL-6 level > 412 U/mL, CCI > 3, and critical disease were obtained to predict ICU admission, ventilator use, and ECMO use. Initial KL-6 level > 322 U/mL was applied to predict 30-day mortality. Kaplan–Meier survival analysis was used to estimate the HR with a 95% confidence interval (CI) for 30-day mortality for age > 70 years, initial KL-6 level > 322 U/mL, CCI > 3, and critical disease. The sample size for the Kaplan–Meier survival analysis was estimated based on our previous study, and the inputs were identical to those described in our previous study, except for the alternative survival probability [2]. The alternative survival probability was set to set to *S*_1_(*t*) = 0.167 based on the 30-day mortality of this study. The sample size of 48 was considered sufficient to perform the Kaplan–Meier survival analysis. MedCalc Software (version 20.111, MedCalc Software, Ostend, Belgium) was used for statistical analysis. *p* value < 0.05 was considered statistically significant.

## 3. Results

Basic characteristics of the study population are summarized in Table 1. The median age (IQR) was 72.0 years (63.0–79.0), and males were 58.4% (*n* = 23). Among four patients with pulmonary disease, one patient had ILD with ARDS and sepsis. In laboratory data, white blood cells (WBC), neutrophils, total bilirubin, lactate dehydrogenase (LDH), and C-reactive protein (CRP) levels differed significantly according to initial KL-6 quartiles (all *p* < 0.05). In the severity assessment, the SOFA score differed significantly according to initial KL-6 quartiles (*p* = 0.005). The distribution of WHO disease severity did not differ significantly according to initial KL-6 quartiles, but the distribution of ARDS and septic shock differed significantly according to initial KL-6 quartiles. In clinical outcomes, ICU admission, ventilator use, and ECMO use differed significantly according to initial KL-6 quartiles, but not the 30-day mortality.

Table 2 shows the comparison of age, initial KL-6 level, CCI, and critical disease according to clinical outcomes. The proportion of age > 70 years differed significantly according to the 30-day mortality; there was no significant difference according to ICU admission, ventilator use, and ECMO use. The proportion of initial KL-6 level > 322 U/mL (or >412 U/mL) differed significantly according to ICU admission, ventilator use, and ECMO use, except for the 30-day mortality. The proportion of CCI > 3 did not differ significantly according to all clinical outcomes. The proportion of critical disease differed significantly according to ventilator use.

KL-6 levels according to WHO disease severity and 30-day mortality are presented in Figure 2. Median initial KL-6 level was higher in critical disease than in moderate disease, showing no statistical significance (251.0 U/mL [179.3–513.0] vs. 174.0 U/mL [131.5–297.0]). In critical disease, the median initial KL-6 level was significantly higher in septic shock than in sepsis (565.5 U/mL [474.0–778.0] vs. 222.0 U/mL [163.5–420.8], *p* = 0.008) (Figure 2A). There were no significant differences between the initial and F/U KL-6 levels in both survivors and non-survivors (initial KL-6 vs. F/U KL-6: 214.5 U/mL [160.0–367.0] vs. 217.0 U/mL [140.5–261.5] in survivors, 432.0 U/mL [195.5–814.5] vs. 622.0 U/mL [311.0–857.5] in non-survivors) (Figure 2B). Median F/U KL-6 level was significantly higher in non-survivors than in survivors (622.0 U/mL [311.0–857.5] vs. 217.0 [140.5–261.5], *p* = 0.003).

In the ROC curve analysis, initial KL-6 level > 412 U/mL predicted ICU admission, ventilator use, and ECMO use significantly better than age > 70 years, CCI > 3, and critical disease (all *p* < 0.05) (Figure 3). Initial KL-6 level > 322 U/mL predicted the 30-day mortality comparably with age > 70 years, CCI > 3, and critical disease (all *p* > 0.05). In the Kaplan–Meier survival analysis, HR (95% CI) for predicting the 30-day mortality was 4.8 (1.2–19.3) for age > 70 years, 4.7 (1.1–21.6) for initial KL-6 level > 322 U/mL, 3.9 (0.9–16.2) for CCI > 3, and 2.1 (0.5–10.3) for critical disease (Figure 4).

## 4. Discussion

This is the first study that explored KL-6 levels, age, CCI, and critical disease simultaneously in hospitalized COVID-19 patients. Our data showed that the KL-6 level reflected COVID-19 disease severity. Initial KL-6 quartiles showed a stepwise increase in critical disease, ARDS, and septic shock (Table 1). Initial KL-6 level was higher in critical disease than in moderate disease without significance, but it was significantly higher in septic shock than in sepsis (Figure 2). The present findings are in line with previous findings [20,21,24,26,28].

In this study, most initial KL-6 levels were lower than 500 U/mL, the cut-off value for ILD [18]. Similar to our study, previous studies demonstrated that the median KL-6 level in COVID-19 was lower than 500 U/mL regardless of disease severity or prognosis [20,22,28,29]. In this study, only one 86-year-old female patient had ILD. She was admitted to ICU for ventilator use with ARDS and sepsis at the time of COVID-19 diagnosis. ECMO was not used. Her initial KL-6 level at ICU admission was 1103 U/mL, and it was the second highest level. She died 29 days after admission.

It has been reported that KL-6 levels significantly correlated with other inflammatory biomarkers, such as CRP, neutrophils, and IL-6 levels in COVID-19 [8,27]. KL-6 levels could increase in ARDS and could be affected by alveolar epithelial damage and the activity of TNF-α [18,20,25,29,32]. If COVID-19-induced ARDS persists for longer than three weeks, it can develop into PF [13]. In this study, eight patients presented with COVID-19-induced ARDS for longer than three weeks, but none of them underwent F/U chest tomography. Therefore, we could not confirm whether post-COVID-19 PF developed or not in these patients. Further studies on the relationship among KL-6, conventional inflammatory biomarkers, or organ-specific biomarkers are needed to assess the mechanism of the increased KL-6 levels in severe/critical COVID-19.

This study showed that initial KL-6 level was significantly associated with ICU admission, ventilator use, and ECMO use (Table 1 and Table 2 and Figure 2). Regarding 30-day mortality, non-survivors had a higher proportion of initial KL-6 levels above the optimal cut-off value without significance. Age > 70 years was only significantly associated with 30-day mortality. CCI > 3 was not associated with all clinical outcomes. Critical disease was only significantly associated with ventilator use. The number of ICU admissions, ventilator use, ECMO use, and 30-day mortality was the highest in Q4 (initial KL-6 level ≥ 449.0 U/mL). Each ICU admission, ventilator use, ECMO use, and 30-day mortality rate had a higher proportion of initial KL-6 levels above their respective cut-off values than each control group. Both initial and F/U KL-6 levels were higher in non-survivors than in survivors. In addition, this study showed a substantial increase in the median KL-6 level in non-survivors in serial KL-6 measurements. Although the time points of serial KL-6 level measurements varied across previous studies [22,25,29,30], based on our data and previous studies, KL-6 levels seem to reflect COVID-19 disease progression.

Initial KL-6 level significantly predicted ICU admission, ventilator use, and ECMO use better than age, CCI, and critical disease (Figure 3). Although age and CCI were the only significant predictors of 30-day mortality, the predictive performance of KL-6 was comparable to that of age and CCI. HRs of both age and initial KL-6 level were significantly high, which were almost the same. COVID-19 prognosis is unpredictable, and patients could progress rapidly to a deteriorating state requiring critical care [3,15]. In Japan, COVID-19 patients with critical care demand accounted for 106.3% of designated medical institutions during the peak period of the COVID-19 epidemic [35]. Patients requiring a ventilator or ECMO were 88.9% and 17.7% of designated medical institutions, respectively [35]. Therefore, it is important for hospitals to have both an early prediction system and clinical care capacity for prompt management [3,35,36]. The WHO recommends monitoring signs of clinical deterioration with vital signs, clinical scores, laboratory data, electrocardiogram, or chest imaging [3]. If an ideal biomarker is available, that would be a simpler and more objective parameter compared with the aforementioned signs. Based on our data, KL-6 could be utilized as a parameter for the risk assessment requiring ICU, ventilator, or ECMO, and for predicting 30-day survival. In this study, most ARDS patients received ventilator and ECMO therapy. KL-6 may be a parameter for a tailored approach or prognostication in COVID-19-induced ARDS.

This study is limited in that it was a small-sized, single-center study that was conducted during the early COVID-19 pandemic. Although the sample size (*n* = 48) was enough for statistical analyses, it may not be enough to have strong implications for clinical insights into the usefulness of KL-6. In addition, our data may have been affected by the study population being biased toward critical diseases and may not be representative of COVID-19 by other variants. Second, we could not conduct blood sampling at a defined exact time, and thus, our data may have been influenced by the clinical course of COVID-19 [2]. Third, we assessed KL-6 levels only to predict ICU admission, ventilator use, ECMO use, and 30-day mortality. The prediction of post-COVID-19 PF was not within the scope of this study due to a lack of information on it.

In conclusion, this is the first study that investigated the usefulness of KL-6 for predicting clinical outcomes in hospitalized COVID-19 patients compared with age, CCI, and critical disease. KL-6 was superior to age, CCI, and critical disease for predicting ICU admission, ventilator use, and ECMO use; for predicting 30-day mortality, they showed comparable performance. KL-6 could be an objective biomarker for predicting COVID-19 patients with critical care demand. Further studies are needed to implement KL-6 as a predictive biomarker for COVID-19 in routine clinical practice. Despite these limitations, this study provides basic data on the predictive power of KL-6 for future pandemics of respiratory viruses other than COVID-19.

## Figures and Tables

**Figure 1 medicina-58-01317-f001:**
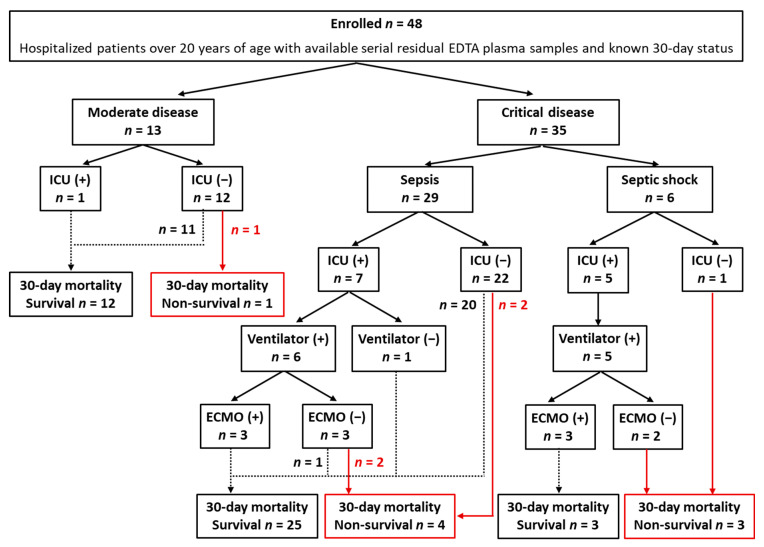
The diagram of enrollment and clinical outcomes of the study population. None of the study population presented mild or severe disease of WHO disease severity. Dotted and red lines indicate survival and death with regard to 30-day mortality, respectively. Abbreviations: EDTA, ethylene-diamine-tetraacetic acid; ICU, intensive care unit; ECMO, extracorporeal membrane oxygenation.

**Figure 2 medicina-58-01317-f002:**
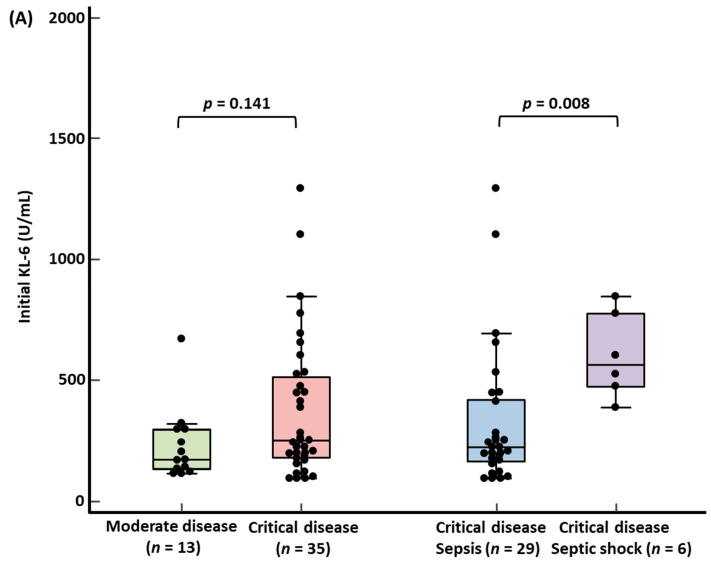
KL-6 levels according to WHO disease severity and 30-day mortality. (**A**) Initial KL-6 levels according to WHO disease severity. (**B**) Initial and F/U KL-6 levels according to 30-day mortality. * Wilcoxon signed-rank test between initial and F/U KL-6 levels. ^†^ Mann–Whitney test for initial KL-6 levels between survivors and non-survivors. ^‡^ Mann–Whitney test for F/U KL-6 levels between survivors and non-survivors. Abbreviations: KL-6, Krebs von den Lungen-6; and F/U, follow-up.

**Figure 3 medicina-58-01317-f003:**
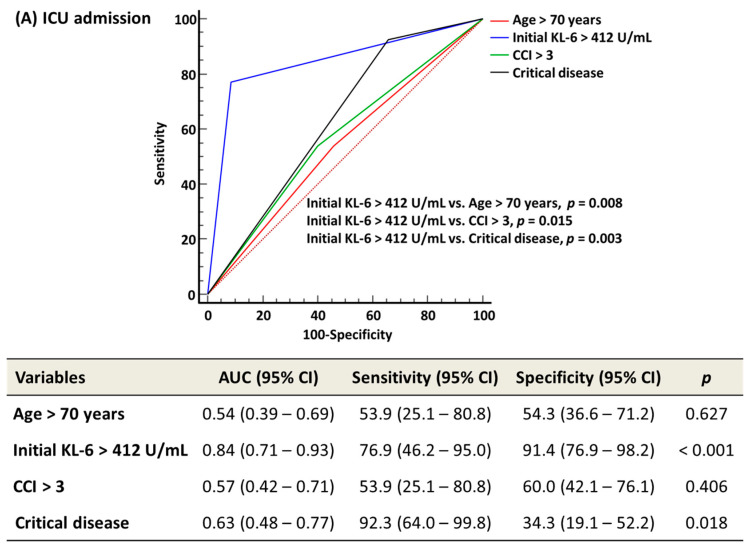
The ROC curve analysis of age, initial KL-6 level, CCI, and critical disease to predict clinical outcomes. In all variables except initial KL-6 level, the optimal cut-off values for 30-day mortality were applied. In initial KL-6, the respective optimal cut-off values were applied; 412 U/mL for ICU admission, ventilator use, and ECMO use and 322 U/mL for 30-day mortality. (**A**) ICU admission. (**B**) Ventilator use. (**C**) ECMO use. (**D**) 30-day mortality. Abbreviations: ROC; receivers operating characteristic; KL-6, Krebs von den Lungen-6; CCI, Charlson comorbidity index; ICU, intensive care unit; ECMO, extracorporeal membrane oxygenation; AUC, area under the curve; and CI, confidence interval.

**Figure 4 medicina-58-01317-f004:**
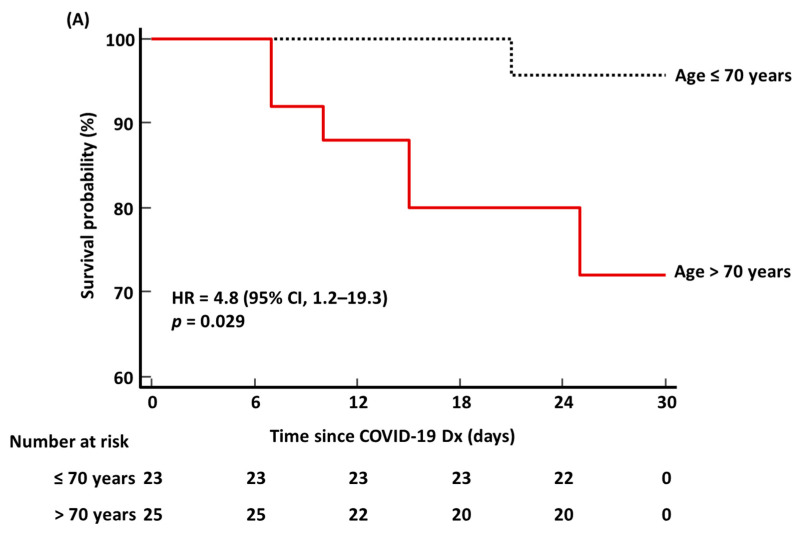
Kaplan–Meier survival analysis of age, initial KL-6 level, CCI, and critical disease for 30-day mortality. (**A**) Age. (**B**) Initial KL-6 level. (**C**) CCI. (**D**) Critical disease. Abbreviations: KL-6, Krebs von den Lungen-6; CCI, Charlson comorbidity index; HR, hazard ratio; CI, confidence interval; Dx, diagnosis.

**Table 1 medicina-58-01317-t001:** Basic characteristics of the study population.

Variable	Total (*n* = 48)	Q1 (*n* = 12)	Q2 (*n* = 12)	Q3 (*n* = 12)	Q4 (*n* = 12)	*p* *
Age, years	72.0 (63.0–79.0)	68.0 (57.0–74.5)	73.5 (63.0–83.0)	75.0 (70.0–77.0)	71.0 (64.5–79.5)	0.658
Male	28 (58.3)	7 (58.3)	9 (75.0)	6 (50.0)	6 (50.0)	0.561
Comorbidities						
HTN	25 (52.1)	6 (50.0)	5 (41.7)	6 (50.0)	8 (66.7)	0.663
DM	16 (33.3)	4 (33.3)	3 (25.0)	3 (25.0)	6 (50.0)	0.522
Solid cancer	7 (14.6)	1 (8.3)	3 (25.0)	2 (16.7)	1 (8.3)	0.606
Pulmonary disease ^†^	4 (8.3)	1 (8.3)	0 (0.0)	1 (8.3)	2 (16.7)	0.350
CHF	4 (8.3)	1 (8.3)	2 (16.7)	1 (8.3)	0 (0.0)	0.535
Dementia	3 (6.2)	1 (8.3)	0 (0.0)	2 (16.7)	0 (0.0)	0.270
CVD	3 (6.2)	2 (16.7)	1 (8.3)	0 (0.0)	0 (0.0)	0.271
CAD	3 (6.2)	1 (8.3)	1 (8.3)	0 (0.0)	1 (8.3)	0.785
CKD	2 (4.2)	0 (0.0)	1 (8.3)	0 (0.0)	1 (8.3)	0.555
PAD	2 (4.2)	0 (0.0)	0 (0.0)	1 (8.3)	1 (8.3)	0.555
Liver disease	2 (4.2)	0 (0.0)	1 (8.3)	1 (8.3)	0 (0.0)	0.555
Rheumatic disease	1 (2.1)	1 (8.3)	0 (0.0)	0 (0.0)	0 (0.0)	0.382
Symptoms						
Fever (≥37.5 °C)	33 (68.7)	7 (58.3)	9 (75.0)	9 (75.0)	8 (66.7)	0.785
Dyspnea	25 (52.1)	5 (41.7)	5 (41.7)	7 (58.3)	8 (66.7)	0.521
General weakness	23 (47.9)	7 (58.3)	6 (50.0)	6 (50.0)	4 (33.3)	0.663
Cough	18 (37.5)	5 (41.7)	6 (50.0)	4 (33.3)	3 (25.0)	0.619
Sputum	11 (22.9)	5 (41.7)	4 (33.3)	1 (8.3)	1 (8.3)	0.111
Fatigue	10 (20.8)	4 (33.3)	3 (25.0)	1 (8.3)	2 (16.7)	0.471
Gastrointestinal ^‡^	6 (12.5)	2 (16.7)	1 (8.3)	2 (16.7)	1 (8.3)	0.858
Chilling	5 (10.4)	2 (16.7)	1 (8.3)	1 (8.3)	1 (8.3)	0.880
Myalgia	5 (10.4)	2 (16.7)	2 (16.7)	0 (0.0)	1 (8.3)	0.483
Headache	5 (10.4)	2 (16.7)	1 (8.3)	2 (16.7)	0 (0.0)	0.483
Sore throat	2 (4.2)	1 (8.3)	0 (0.0)	0 (0.0)	1 (8.3)	0.555
Hemoptysis	1 (2.1)	0 (0.0)	1 (8.3)	0 (0.0)	0 (0.0)	0.382
Rhinorrhea	1 (2.1)	0 (0.0)	0 (0.0)	0 (0.0)	1 (8.3)	0.382
Nasal obstruction	1 (2.1)	1 (8.3)	0 (0.0)	0 (0.0)	0 (0.0)	0.382
Chest pain	1 (2.1)	0 (0.0)	0 (0.0)	0 (0.0)	1 (8.3)	0.382
COVID-19 Dx. to enrollment, days	4.0 (0.5–9.0)	3.5 (0.5–12.0)	3.5 (1.0–8.5)	7.0 (2.0–12.0)	4.5 (0.0–8.0)	0.813
Vital signs						
SBP, mm Hg	130.0 (112.0–142.5)	136.0 (125.0–145.0)	135.5 (122.5–140.0)	120.0 (114.0–160.0)	114.0 (104.0–138.5)	0.294
DBP, mm Hg	77.0 (70.0–80.5)	80.0 (70.0–90.0)	74.0 (70.0–80.0)	74.5 (65.0–80.5)	74.5 (69.5–85.0)	0.834
HR, beats/min	83.5 (76.5–93.0)	84.5 (81.0–93.5)	79.5 (72.0–88.0)	85.5 (76.0–96.5)	83.5 (75.0–98.5)	0.692
RR, breaths/min	20.0 (20.0–23.5)	20.0 (20.0–20.5)	21.0 (20.0–22.5)	20.0 (20.0–22.5)	24.0 (17.0–28.5)	0.397
BT, °C	37.2 (36.9–37.8)	37.0 (36.9–37.9)	37.5 (37.1–37.9)	37.2 (36.8–37.6)	37.0 (37.7)	0.335
SpO_2_, %	95.9 (93.8–97.1)	97.0 (94.4–98.8)	96.0 (95.0–96.9)	94.0 (93.0–97.5)	94.9 (90.0–97.0)	0.481
Laboratory data						
WBC, ×10^9^/L	6.4 (5.0–8.2)	4.9 (4.2–6.1)	6.3 (5.7–8.9)	6.8 (5.1–7.6)	9.1 (6.7–13.1)	0.007
Neutrophils, ×10^9^/L	4.8 (3.2–6.7)	3.1 (2.5–3.8)	4.7 (4.3–5.9)	5.0 (3.6–6.4)	7.9 (5.5–11.5)	0.002
Lymphocytes, ×10^9^/L	1.0 (0.7–1.5)	1.4 (0.8–1.6)	1.0 (0.7–1.5)	1.0 (0.6–1.5)	0.8 (0.6–1.1)	0.268
Hb, g/L	117.0 (109.5–130.5)	117.0 (110.0–132.5)	114.5 (102.5–132.0)	116.0 (101.5–129.5)	122.5 (115.0–130.0)	0.915
PLT, ×10^9^/L	213.0 (156.5–301.5)	234.5 (141.5–322.5)	235.5 (184.0–293.0)	193.5 (156.5–309.5)	193.0 (144.0–269.5)	0.901
AST, IU/L	31.5 (24.5–47.0)	28.0 (23.5–44.5)	28.0 (22.0–32.0)	43.5 (26.5–63.5)	39.5 (30.0–54.0)	0.085
ALT, IU/L	26.0 (15.3–44.0)	25.0 (15.0–46.0)	25.0 (14.0–32.0)	33.0 (17.0–55.0)	37.0 (17.3–46.3)	0.537
ALP, IU/L	72.0 (62.5–89.5)	67.5 (60.0–76.0)	80.5 (72.5–93.5)	69.0 (57.0–124.0)	71.0 (63.0–63.5)	0.399
Total bilirubin, umol/L	12.7 (9.0–17.5)	9.5 (6.2–11.5)	12.3 (8.8–16.8)	12.5 (8.7–13.9)	23.2 (15.6–36.3)	0.002
LDH, IU/L	513.5 (420.5–665.5)	419.5 (338.0–548.0)	457.0 (390.8–514.8)	545.0 (474.5–638.5)	876.0 (697.8–1061.3)	<0.001
BUN, mmol/L	5.7 (3.6–8.0)	3.8 (2.8–5.7)	5.9 (5.4–9.4)	4.8 (3.2–6.8)	7.1 (5.4–9.9)	0.064
Cr, umol/L	68.0 (53.0–91.9)	61.9 (49.5–76.0)	74.3 (57.4–98.6)	65.9 (54.8–83.9)	80.0 (49.1–133.1)	0.372
CRP, mg/L	45.0 (9.0–122.0)	10.0 (2.0–47.0)	22.0 (13.0–116.0)	57.0 (12.0–112.0)	99.0 (63.0–156.0)	0.039
Initial KL-6, U/mL	234.5 (160.0–449.0)	114.0 (98.0–128.0)	198.5 (175.0–206.0)	290.0 (251.5–356.0)	661.5 (530.5–812.0)	< 0.001
Severity assessment						
CCI	4.0 (2.0–5.0)	3.5 (2.0–5.0)	4.0 (2.0–5.5)	4.0 (2.5–5.0)	4.0 (3.0–4.5)	0.951
SOFA	3.0 (1.0–6.0)	2.0 (0.5–3.5)	3.5 (1.5–4.5)	2.5 (1.0–3.5)	9.0 (5.0–11.5)	0.005
WHO disease severity ^§^						
Moderate disease	13 (27.1)	5 (41.7)	3 (25.0)	4 (33.3)	1 (7.7)	0.297
Critical disease ^‖^	35 (72.9)	7 (58.3)	9 (75.0)	8 (66.7)	11 (91.7)	
ARDS	10 (20.8)	0 (0.0)	1 (8.3)	2 (16.7)	7 (58.3)	<0.001
Sepsis	29 (60.4)	7 (58.3)	9 (75.0)	7 (58.3)	6 (50.0)	0.647
Septic shock	6 (12.5)	0 (0.0)	0 (0.0)	1 (8.3)	5 (41.7)	0.005
Clinical outcome						
ICU admission	13 (27.1)	1 (8.3)	2 (16.7)	1 (8.3)	9 (75.0)	<0.001
Ventilator use	11 (22.9)	0 (0.0)	1 (8.3)	1 (8.3)	9 (75.0)	<0.001
ECMO use ^¶^	6 (12.5)	0 (0.0)	0 (0.0)	1 (8.3)	5 (41.7)	0.005
30-day mortality	8 (16.7)	2 (16.7)	0 (0.0)	2 (16.7)	4 (33.3)	0.187
In-hospital mortality	12 (25.0)	2 (16.7)	2 (16.7)	3 (25.0)	5 (41.7)	0.446
Hospital stay, days	28.0 (21.0–41.0)	24.0 (17.5–29.0)	28.0 (22.0–42.0)	40.0 (26.0–53.5)	40.0 (26.0–53.5)	0.196

Data were presented as a number (percentage) or median (interquartile range). KL-6 levels were divided into quartiles: Q1 < 160.0 U/mL; 160.0 U/mL ≤ Q2 < 234.5 U/mL; 234.5 U/mL ≤ Q3 < 449.0 U/mL; and Q4 ≥ 449.0 U/mL. * *p* values were calculated using the Kruskal–Wallis test or chi-squared test among KL-6 quartile groups. ^†^ COPD (N = 2), asthma (N = 1), and ILD (N =1). ^‡^ Gastrointestinal symptoms included: anorexia, abdominal pain, abdominal distension, or diarrhea. ^§^ None of the study population presented mild or severe disease of WHO disease severity. ^‖^ In critical disease, six sepsis patients and four septic shock patients co-presented ARDS. One sepsis patient co-presented acute thrombosis. ^¶^ In eleven ventilated patients, six patients were treated with ECMO. Abbreviations: HTN, hypertension; DM, diabetes mellitus; CHF, congestive heart failure; CVD, cerebrovascular disease; CAD, cardiovascular disease; CKD, chronic kidney disease; PAD, peripheral artery disease; COVID-19, Coronavirus disease 2019; Dx, diagnosis; SBP, systolic blood pressure; DBP, diastolic blood pressure; HR, heart rate; RR, respiratory rate; BT, body temperature; SpO_2_, saturation of percutaneous oxygen; WBC, white blood cells; Hb, hemoglobin; PLT, platelet; AST, aspartate aminotransferase; ALT, alanine aminotransferase; ALP, alkaline phosphatase; LDH, lactate dehydrogenase; BUN, blood urea nitrogen; Cr, creatinine; CRP, C-reactive protein; KL-6, Krebs von den Lungen-6; CCI, Charlson comorbidity index; SOFA, sequential organ failure assessment; WHO, World Health Organization; ICU, intensive care unit; ECMO, extracorporeal membrane oxygenation; ARDS, acute respiratory distress syndrome; COPD, chronic obstructive pulmonary disease; and ILD, interstitial lung disease.

**Table 2 medicina-58-01317-t002:** Age, initial KL-6 level, CCI, and critical disease according to clinical outcomes.

Variables *	ICU Admission			Ventilator Use			ECMO Use			30-Day Mortality		
	Yes (*n* = 13)	No (*n* = 35)	*p*	Yes (*n* = 11)	No (*n* = 37)	*p*	Yes (*n* = 6)	No (*n* = 42)	*p*	Non-Survival (*n* = 8)	Survival (*n* = 40)	*p*
Age > 70 years	6 (46.2)	19 (54.3)	0.620	5 (45.5)	20 (54.1)	0.619	2 (33.3)	23 (54.8)	0.407	7 (87.5)	18 (45.0)	0.049
Initial KL-6 level												
>322 U/mL	10 (76.9)	5 (14.3)	<0.001	10 (90.9)	5 (14.3)	<0.001	6 (100)	9 (21.4)	<0.001	5 (62.5)	10 (25.0)	0.087
>412 U/mL ^†^	10 (76.9)	3 (8.6)	<0.001	10 (90.9)	3 (8.1)	<0.001	6 (100)	7 (16.7)	<0.001	4 (50.0)	9 (22.5)	0.187
CCI > 3	6 (46.2)	21 (60.0)	0.395	5 (45.5)	22 (59.5)	0.416	2 (33.3)	25 (59.5)	0.383	7 (87.5)	20 (50.0)	0.064
Critical disease	12 (92.3)	23 (65.7)	0.081	11 (100.0)	24 (64.9)	0.023	6 (100.0)	29 (69.0)	0.171	7 (87.5)	28 (70.0)	0.418

Data were presented as a number (percentage). * The optimal cut-off values of variables for 30-day mortality were applied; age > 70 years, initial KL-6 level > 322 U/mL, CCI > 3, and critical disease. ^†^ The optimal cut-off value of initial KL-6 for ICU admission, ventilator use, and ECMO use was 412 U/mL, which was the same for all. Abbreviations: KL-6, Krebs von den Lungen-6; CCI, Charlson comorbidity index; ICU, intensive care unit; and ECMO, extracorporeal membrane oxygenation.

## Data Availability

The data presented in this study are available on request from the corresponding author.

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
