# Peer review of "Usefulness of KL-6 for Predicting Clinical Outcomes in Hospitalized COVID-19 Patients"

_medicina, 2022, doi:10.3390/medicina58101317_

Round 1
Reviewer 1 Report
Thanks for your labor-intensive study.
The manuscript with the title “Usefulness of KL-6 for Predicting Clinical Outcomes in Hospitalized COVID-19 Patients” has been reviewed by us now. The authours investigated usefulness of Krebs von den Lungen 6 (KL-6), a novel biomarker for interstitial lung disease, to predict clinical outcomes in hospitalized coronavirus disease 2019 (COVID-19). They evaluated 48 hospitalized COVID-19 patients and measured KL-6 with an automated analyzer. Also, they recorded clinical outcomes of the patients and analyzed their relationship with KL-6 level by statistical analysis. As a result, they suggested that KL-6 level may have important beneficial effects such as the need for intensive care unit, the need for ventilators, and mortality pretiction in COVID-19 patients. These findings were clear and compatible with the literature. Therefore, the paper presents an innovation and contributes to the literature. Also, the design of the study and the writing of the paper are also convenient and fine. Thanks the authours for their labour-intensive study.
Best regards.
Associate Prof. Cuma MERTOGLU Ph.D, M.D
Author Response
Reviewer #1
The manuscript with the title “Usefulness of KL-6 for Predicting Clinical Outcomes in Hospitalized COVID-19 Patients” has been reviewed by us now. The authours investigated usefulness of Krebs von den Lungen 6 (KL-6), a novel biomarker for interstitial lung disease, to predict clinical outcomes in hospitalized coronavirus disease 2019 (COVID-19). They evaluated 48 hospitalized COVID-19 patients and measured KL-6 with an automated analyzer. Also, they recorded clinical outcomes of the patients and analyzed their relationship with KL-6 level by statistical analysis. As a result, they suggested that KL-6 level may have important beneficial effects such as the need for intensive care unit, the need for ventilators, and mortality pretiction in COVID-19 patients. These findings were clear and compatible with the literature. Therefore, the paper presents an innovation and contributes to the literature. Also, the design of the study and the writing of the paper are also convenient and fine. Thanks the authours for their labour-intensive study.
Thank you very much for your comprehensive and kind review.
Reviewer 2 Report
The authors have presented a biomarker study on COVID-19 patients exploring the performance of using KL-6.
I have the following concerns about this work:
1. The sample size is very low (only 48 patients), I do not expect any strong implications of the results for a clinical insight.
+
2. Authors need to provide more evidence on why they selected KL-6 to be syudied as abiomarker for COVID-19.
3. English language of the paper needs improvement.
Author Response
Reviewer #2
The authors have presented a biomarker study on COVID-19 patients exploring the performance of using KL-6. I have the following concerns about this work:
1. The sample size is very low (only 48 patients), I do not expect any strong implications of the results for a clinical insight.
Thank you for your comment. According to your comment, we modified the following sentences in Discussion section.
Although the sample size (N = 48) was enough for statistical analyses, it may not be enough to make strong implications for clinical insight on the usefulness of KL-6. In addition, our data may have been affected by the study population being biased toward critical disease and could not be representative of COVID-19 by other variants. (page 12, line 237).
2. Authors need to provide more evidence on why they selected KL-6 to be syudied a biomarker for COVID-19.
Thank you for your comment. According to your comment, we added or modified the following sentences in Introduction section. We also added new references.
Recent studies have explored the association between KL-6 levels and COVID-19 disease severity or prognosis; KL-6 levels increased according to COVID-19 disease severity and significantly predicted hospital days and poor prognosis in some studies [20-32]. KL-6 has been also suggested as a potential biomarker for post-COVID-19 PF [7]. KL-6 was significantly associated with radiological abnormalities after COVID-19 and predicted post-COVID-19 PF [22, 30].
To the best of our knowledge, no studies have evaluated the predictive performance of KL-6 compared with age, comorbidities, and disease severity in COVID-19 simultaneously. We hypothesized that KL-6 might be beneficial in predicting clinical outcomes in COVID-19. We explored the usefulness of KL-6 for predicting clinical outcomes in hospitalized COVID-19 patients compared with age, CCI, and WHO disease severity. We also analyzed serial KL-6 levels according to the 30-day mortality. (page 4, line 71).
31. D'Agnano, V.; Scialo, F.; Perna, F.; Atripaldi, L.; Sanduzzi, S.; Allocca, V.; Vitale, M.; Pastore, L.; Bianco, A.; Perrotta, F. Exploring the Role of Krebs von den Lungen-6 in Severe to Critical COVID-19 Patients. Life (Basel) 2022, 12, 1141.
32. Castellví, I.; Castillo, D.; Corominas, H.; Mariscal, A.; Orozco, S.; Benito, N.; Pomar, V.; Baucells, A.; Mur, I.; de la Rosa-Carrillo, D.; et al. Krebs von Den Lungen-6 Glycoprotein Circulating Levels Are Not Useful as Prognostic Marker in COVID-19 Pneumonia: A Large Prospective Cohort Study. Front Med (Lausanne) 2022, 9, 973918.
3. English language of the paper needs improvement.
Thank you for your comment. According to your comment, we checked typos or grammatical errors throughout the whole manuscript and trimmed the sentences.
Reviewer 3 Report
In the face of a new disease, medical researchers make sustained efforts to discover specific biomarkers for the diagnosis or prediction of its evolution. The authors explore the usefulness of the KL-6 marker for the severe evolution of COVID 19, i.e. patients at risk of ICU, ventilator use and ECMO required. The study design is well specified. The title corresponds to the content of the article. The authors perform a statistical analysis following the sensitivity and specificity of this biomarker for each of the severity criteria. The study provides arguments for the usefulness of KL-6 for predicting clinical outcome in patients hospitalized for COVID, given that previous studies had contradictory results.
My question would be what is the clinical utility of this investigation? Could it influence the therapeutic decision? As this analysis of KL-6 is not very accessible, does it correlate with the usual inflammatory markers (BRS, fibrinogen, CRP, IL 6, etc.)? Moreover, can the authors explain what is the mechanism by which the increase of KL-6 in severe forms of COVID occurs more than the insult at the level of type II pneumocytes?
Author Response
Reviewer #3
In the face of a new disease, medical researchers make sustained efforts to discover specific biomarkers for the diagnosis or prediction of its evolution. The authors explore the usefulness of the KL-6 marker for the severe evolution of COVID 19, i.e. patients at risk of ICU, ventilator use and ECMO required. The study design is well specified. The title corresponds to the content of the article. The authors perform a statistical analysis following the sensitivity and specificity of this biomarker for each of the severity criteria. The study provides arguments for the usefulness of KL-6 for predicting clinical outcome in patients hospitalized for COVID, given that previous studies had contradictory results.
1. My question would be what is the clinical utility of this investigation? Could it influence the therapeutic decision?
Thank you for your comment. According to your comment, we modified the following sentences in Methods and Discussion sections.
Among them, ventilators were applied to 11 patients (ARDS [N = 9]), and ECMO was applied to six ventilated ARDS patients. 30-day mortality was 16.7% (N = 8). (page 5, line 100)
Based on our data, KL-6 could be utilized as a parameter for the risk assessment requiring ICU, ventilator, or ECMO and for predicting 30-day survival. In this study, most ARDS patients received ventilator and ECMO therapy. KL-6 may be a parameter for a tailored approach or prognostication in COVID-19-induced ARDS. (page 12, line 232)
2. As this analysis of KL-6 is not very accessible, does it correlate with the usual inflammatory markers (BRS, fibrinogen, CRP, IL 6, etc.)? Moreover, can the authors explain what is the mechanism by which the increase of KL-6 in severe forms of COVID occurs more than the insult at the level of type II pneumocytes?
Thank you for your comment. According to your comment, we added or modified the following sentences in Introduction and Discussion sections.
When SARS-CoV-2 enters host cells via ACE2, angiotensin (Ang) â…¡ is produced, followed by the production of proinflammatory cytokines such as tumor necrosis factor alpha (TNFα) and interlukin-6 (IL-6) [8-12]. It results in inflammation, fibrosis, lung damage, and edema. (page 3, line 57)
KL-6 is a novel biomarker for ILD, and it is produced by injured/regenerating type â…¡ pneumocytes as well as bronchial epithelial cells and other cells [7, 18, 19]. (page 4, line 70)
It has been reported that KL-6 level significantly correlated with other inflammatory biomarkers such as CRP, neurophils, and IL-6 levels in COVID-19 [8, 27]. KL-6 level could increase in ARDS and could be affected by alveolar epithelial damage and the activity of TNFα [18, 20, 25, 29, 32]. If COVID-19-induced ARDS persists longer than three weeks, it can develop into PF [13]. In this study, eight patients presented with COVID-19-induced ARDS for longer than three weeks, but none of them underwent F/U chest tomography. Therefore, we could not confirm whether post-COVID-19 PF developed or not in these patients. Further studies on the relationship among KL-6, conventional inflammatory biomarkers, or organ-specific biomarkers are needed to assess the mechanism of the increased KL-6 level in severe/critical COVID-19. (page 10, line 195)
Round 2
Reviewer 2 Report
The authors have addressed all my concerns and I recommewnd this version for publishing.